# Interpretable Chirality-Aware Graph Neural Network for Quantitative Structure Activity Relationship Modeling

**Yunchao "Lance" Liu**
Vanderbilt University
yunchao.liu@vanderbilt.edu

**Yu Wang**
Vanderbilt University
yu.wang.1@vanderbilt.edu

**Oanh Vu**
Vanderbilt University
oanh.t.vu@vanderbilt.edu

**Rocco Moretti**
Vanderbilt University
rocco.moretti@vanderbilt.edu

**Bobby Bodenheimer**
Vanderbilt University
bobby.bodenheimer@vanderbilt.edu

**Jens Meiler**
Vanderbilt University
Leipzig University
jens.meiler@vanderbilt.edu

**Tyler Derr**
Vanderbilt University
tyler.derr@vanderbilt.edu

## Abstract

In computer-aided drug discovery, quantitative structure activity relation models are trained to predict biological activity from chemical structure. Despite the recent success of applying graph neural networks to this task, important chemical information such as molecular chirality is ignored. To fill this crucial gap, we propose Molecular-Kernel Graph Neural Network (MolKGNN) for molecular representation learning, which features conformation invariance, chirality-awareness, and interpretability. For MolKGNN, we first design a molecular graph convolution to capture the chemical pattern by comparing the atom's similarity with learnable molecular kernels. Furthermore, we propagate the similarity score to capture the higher-order chemical pattern. To assess the method, we conduct a comprehensive evaluation with nine well-curated datasets spanning numerous important drug targets that feature realistically high class imbalance. Meanwhile, the learned kernels identify patterns that agree with domain knowledge, confirming MolKGNN's pragmatic interpretability.

## 1 Introduction

Developing new drugs is time-consuming and expensive, e.g., it took cabozantinib, an oncologic drug, 8.8 years and $1.9 billion to get on the market [1]. To assist this process, computer-aided drug discovery (CADD) has been widely used. One branch of CADD constructs Quantitative Structure Activity Relationship (QSAR) models to predict the biological activity of molecules based on their chemical structure [2].

Graph Neural Networks (GNNs) have successfully been applied in many fields. As molecules can be viewed as graphs with atoms as nodes and chemical bonds as edges, GNNs are a logical choice to construct QSAR models [3]. A typical GNN architecture for graph classification begins with an encoder extracting node representations by passing neighborhood information followed by pooling operations that integrate node representations into graph representations, which are fed into a classifier to predict graph classes [4].

Despite the promise of GNN models applied to molecular representation learning, existing GNN models either blindly follow the message passing framework without considering molecular constraints on

Y. Liu et al., Interpretable Chirality-Aware Graph Neural Network for Quantitative Structure Activity Relationship Modeling (Extended Abstract). Presented at the First Learning on Graphs Conference (LoG 2022), Virtual Event, December 9–12, 2022.

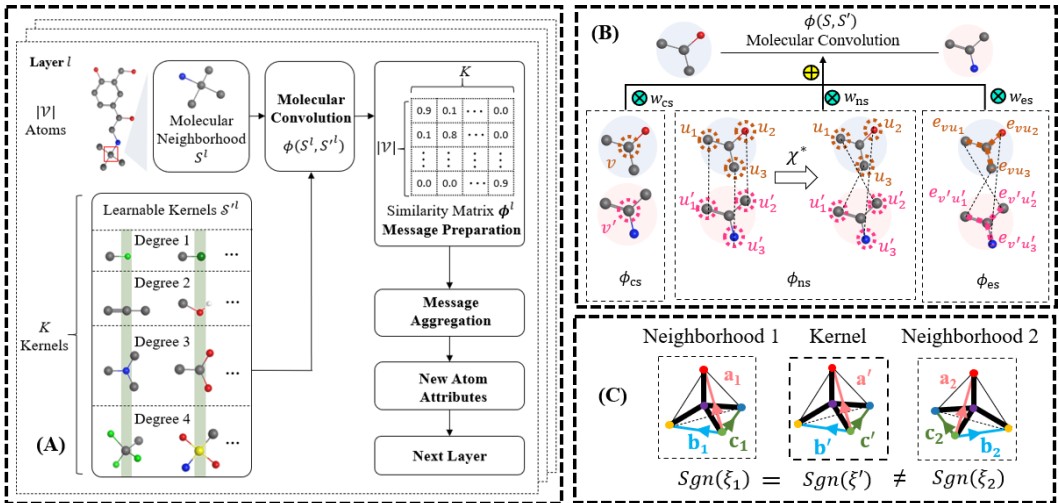

**Figure 1:** (A) An overview of the proposed MolKGNN. (B) An illustration of the molecular convolution that captures three aspects of similarities. (C) An illustration of the chirality calculation.

graphs [5], fail to integrate chirality [6], or lack interpretability [7]. To address these limitations, we develop a GNN model named MolKGNN that features conformation invariance, chirality-awareness and provides a form of interpretability. Our contributions are:

- **Interpretable Molecular Convolution:** We design a new convolution operation to capture chemical pattern of each atom by quantifying the similarity between the atom's neighboring subgraph and the learnable molecular kernel, which is inherently interpretable.

- **Chirality Characterization:** Rather than listing all permutations of neighbors for a chiral center [8], or using dihedral angles [7], the chirality calculation module in MolKGNN uses a lightweight linear algebra calculation.

- **Realistic Benchmark:** We perform a comprehensive evaluation using well-curated datasets spanning numerous important drug targets (that feature realistic high class imbalance) and metrics that bias predicted active molecules for actual experimental validation. Ultimately, we demonstrate the superiority of MolKGNN over other GNNs in CADD.

## 2 Related Work and Preliminaries

Several attempts have been made to leverage GNNs for molecular representation learning. Early models capture the 2D connectivity (i.e., molecular constitution) [9, 10]. However, molecules are not planar but 3D entities and bond lengths/angles/dihedral angles need thus to be taken into considerations [6, 11, 12]. To account for chirality, reflection-sensitive models are designed [8, 13].

In this work, a molecule is represented as an attributed and undirected graph $G = (\mathcal{V}^G, \mathcal{E}^G)$ where $\mathcal{V}^G, \mathcal{E}^G$ are the set of nodes (atoms) and edges (chemical bonds). Let $v \in \mathcal{V}^G$ denote the node $v$ and $e_{vu} \in \mathcal{E}^G$ denote an edge between $v$ and $u$. Moreover, we represent the node attribute matrix as $\mathbf{X}^G \in \mathbb{R}^{|\mathcal{V}^G| \times d_v}$ and edge attribute matrix as $\mathbf{E}^G \in \mathbb{R}^{|\mathcal{V}^G| \times |\mathcal{V}^G| \times d_e}$ where $d_v, d_e$ are the dimension of node and edge features. The node coordinate matrix is represented as $\mathbf{P}^G \in \mathbb{R}^{|\mathcal{V}^G| \times 3}$ and $\mathbf{P}_v^G$ denotes the 3D coordinates of $v$. The graph topology is described by its adjacency matrix $\mathbf{A}^G \in \{1, 0\}^{|\mathcal{V}^G| \times |\mathcal{V}^G|}$ where $\mathbf{A}_{vu}^G = 1$ if $e_{vu} \in \mathcal{E}^G$, and $\mathbf{A}_{vu}^G = 0$ otherwise. Note that bond types are encoded as edge features.

## 3 Molecular-Kernel Graph Neural Network

In this section, we introduce the framework of MolKGNN, shown in Figure 1 (A). Next, we describe our molecular convolution involving three aspects of similarity along with being chirality-aware, and then highlight the entire model architecture.

### 3.1 Molecular Convolution

In 2D images, convolution operation can be regarded as calculating the similarity between the image patch and the image kernel. Larger output values indicate higher visual similarity patterns such as edges, strips, curves [14]. Inspired by that, we design a molecular convolution that outputs higher values when a molecular neighborhood and kernels are more chemically similar.

However, performing convolution on irregular neighborhood subgraphs requires the learnable molecular kernels to have correspondingly different geometrical structures, which is computationally prohibitive. To handle this challenge, for each atom $v$ of degree $d$ in $G$, we only consider its 1-hop star-like neighborhood subgraph $S = (\mathcal{V}^S, \mathcal{E}^S)$ where $\mathcal{V}^S = \{v\} \cup \mathcal{N}_v^G$ and $\mathcal{E}^S = \{e_{vu} | u \in \mathcal{N}_v^G\}$. To make the molecular convolution feasible, we initialize the molecular kernel to also follow star-structure and denote it as $S' = (\mathcal{V}^{S'}, \mathcal{E}^{S'})$ where $\mathcal{V}^{S'} = \{v'\} \cup \mathcal{N}_{v'}^{S'}$ with $v'$ being the central node without loss of generality and $\mathcal{E}^{S'} = \{e_{v'u'} | u' \in \mathcal{N}_{v'}^{S'}\}$. Let the learnable feature matrix and edge feature matrix of $S'$ be $\mathbf{X}^{S'} \in \mathbb{R}^{(d+1) \times d_n}$ and $\mathbf{E}^{S'} \in \mathbb{R}^{d \times d_e}$, respectively.

Then we define the operation of molecular convolution between the atom $v$ and the molecular kernel $S'$ as quantifying the similarity $\phi$ between $v$'s neighborhood subgraph $S$ and the kernel $S'$: $\phi(S, S') = w_{cs}\phi_{cs}(S, S') + w_{ns}\phi_{ns}(S, S') + w_{es}\phi_{es}(S, S')$. where $\phi_{cs}, \phi_{ns}, \phi_{es}$ quantify the similarity from three different aspects: the central similarity, neighborhood similarity, and edge similarity. We combine them together with learnable weights $w_{cs}, w_{ns}, w_{es} \in [0, 1]$ after softmax-normalization.

**Central Similarity.** We first capture the chemical property of atom $v$ itself in $S$ by computing its similarity to the central node $v'$ in the kernel $S'$: $\phi_{cs}(S, S') = \text{sim}(\mathbf{X}_v^S, \mathbf{X}_{v'}^{S'})$. where $\mathbf{X}_v^S, \mathbf{X}_{v'}^{S'}$ are attributes of the central atom $v$ in the subgraph $S$ and the central node $v'$ in the kernel $S'$. The $\text{sim}(\cdot, \cdot)$ is the function measuring vector similarity and we use cosine similarity throughout this work.

**Neighboring Node and Edge Similarity**. Besides the central node, the chemical property of an atom is also impacted by its neighborhood context, which motivates us to further quantify the similarity between 1) the neighboring nodes $\mathcal{N}_v^S$ in $S$ and $\mathcal{N}_{v'}^{S'}$ in $S'$, and 2) the edges $\mathcal{E}^S$ and $\mathcal{E}^{S'}$.

Before calculating $\phi_{ns}, \phi_{es}$ between $S$ and $S'$, we face a matching problem. For example, in Figure 1(B), the node $u_1$ in $S$ has more than one matching candidates, i.e., $\{u_1', u_2', u_3'\}$ in $S'$. Here we seek a bijective matching $\chi^* : \mathcal{N}_v^S \to \mathcal{N}_{v'}^{S'}$ such that the average attribute similarity between $u \in \mathcal{N}_v^S$ and $\chi^*(u) \in \mathcal{N}_{v'}^{S'}$ over all neighbors can be maximized: $\chi^* = \arg\max_\chi \frac{1}{|\mathcal{N}_v^S|} \sum_{u \in \mathcal{N}_v^S} \text{sim}(\mathbf{X}_u^S, \mathbf{X}_{\chi(u)}^{S'})$. Given that exhausting all $|\mathcal{N}_v^S|!$ possible matchings to find the optimal one is computationally infeasible, we significantly simplify this computation by constraining the searching space according to the inherent structure of molecules, which are: 1) node degrees in drug-like molecule graphs are usually less than 5, with most atoms having a degree of 1 and few nodes having a degree of 4 [15]; 2) for nodes of degree 4, only 12 among the total 24 possible matchings are valid after considering chirality [8]. After the node matching, the bijective edge matching is defined as: $\chi^{e,*} : \mathcal{E}^S \to \mathcal{E}^{S'}$ such that the edge $e_{vu} \in \mathcal{E}^S$ if and only if $e_{v'\chi^*(u)} \in \mathcal{E}^{S'}$. Then, we compute $\phi_{ns}$ and $\phi_{es}$ as: $\phi_{ns} = \frac{1}{|\mathcal{N}_v^S|} \sum_{u \in \mathcal{N}_v^S} \text{sim}(\mathbf{X}_u^S, \mathbf{X}_{\chi^*(u)}^{S'})$ and $\phi_{es} = \frac{1}{|\mathcal{N}_v^S|} \sum_{u \in \mathcal{N}_v^S} \text{sim}(\mathbf{E}_{vu}^S, \mathbf{E}_{v'\chi^{e,*}(u)}^{S'})$.

**Chirality Characterization.** Chirality is a key determinant of a molecule's biological activity [16], but only exists when the central atom has four unique neighboring substructures. Given the neighborhood subgraph of an atom $S$ forming the tetrahedron shown in Figure 1 (C) where the four unique neighboring atoms are $\mathcal{N}_v^S = \{u_1, u_2, u_3, u_4\}$, we select $u_1$ without loss of generality as the anchor neighbor to define the three concurrent sides of the tetrahedron $\mathbf{a}^S = \mathbf{P}_{u_2}^S - \mathbf{P}_{u_1}^S, \mathbf{b}^S = \mathbf{P}_{u_3}^S - \mathbf{P}_{u_1}^S, \mathbf{c}^S = \mathbf{P}_{u_4}^S - \mathbf{P}_{u_1}^S$ and further calculate the tetrahedral volume of $S$ as: $\xi^S = \frac{1}{6}*\mathbf{a}^S \times \mathbf{b}^S \cdot \mathbf{c}^S$ Similarly, we calculate $\xi^{S'}$ for the kernel $S'$. Notice, that the sign of the tetrahedron volume of the molecule $\xi^S$ defines its vertices ordering [16].

The simliarity $\phi(S, S')$ is then updated with chirality as $\phi(S, S') = \big(\text{sgn}(\xi^S)\text{sgn}(\xi^{S'})\big)\phi(S, S')$ with $\text{sgn}(\cdot)$ being the sign function.

## 3.2 Model Architecture

Suppose the set of $K$ kernels at layer $l$ be $\mathcal{S}'^l = \{S'^l_k\}^K_{k=1}$, the proposed molecular convolution is applied with the molecular kernel $S'^l_k \in \mathcal{S}'^l$ over the node representation $\mathbf{H}^{l-1}$ at the previous layer $l-1$ to obtain the node similarity matrix at layer $l$ as $\mathbf{\Phi}^l \in \mathbb{R}^{|\mathcal{V}| \times K}$, where $\mathbf{\Phi}^l_{ik} = \phi(S^{l-1}_{v_i}, S'^{l-1}_k)$ defines the similarity between the neighborhood subgraph around the atom $v_i$ and the $k^{\text{th}}$ kernel at layer $l-1$. We note that $\phi(S^{l-1}_{v_i}, S'^{l-1}_k)$ is set to 0 if $S^{l-1}_{v_i}$ and $S'^{l-1}_k$ have different degrees so that back-propagation keeps the parameters in kernels of different degree untouched. The new node representation $\mathbf{H}^l = \mathbf{A}\mathbf{\Phi}^l$. After recursively alternating between the molecular convolution and the message-passing $L$ layers, the final atom representation $\mathbf{H}^L$ describes the chemical pattern up to $L$ hops away of each atom. Molecular representation $\mathbf{G}$ is obtained via global-sum. Ultimately, graph classification is performed using $\hat{\mathbf{Y}} = \sigma f(\mathbf{G})$ with classifier $f(.)$, e.g., Multi-Layer Perceptron, and softmax normalization $\sigma$. Computational complexities for MolKGNN is given in Appendix A.6.

# 4 Experiments

## 4.1 Experimental Settings

**A Realistic Drug Discovery Scenario.** We benchmark MolKGNN in its predictive ability to binary classification of active/inactive. Models are trained on High-Throughput Screening (HTS) results to screen molecules virtually and prioritize acquisition [17]. HTS datasets are of large sizes, have high label imbalance (many more inactive molecules) and often contain false positives [18]. Moreover, an evaluation metric that biases towards molecules with the highest predicted activities is of interest as only these will be acquired or synthesized and tested.

**Datasets.** Well-curated datasets used are from [19, 20]. Details can be found in Appendix A.1.

**Baselines.** *SchNet* [6], *DimeNet++* [21], *SphereNet* [13], *ChIRo* [7] and *KerGNN* [5] are used. The first four are GNNs for molecular representation learning and the last one is a GNN that is architecturally similar to ours. Further details introducing the baselines is provided in Appendix A.11.

**Evaluation Metrics.** Two metrics are used, detailed in Appendix A.10: Logarithmic Receiver-Operating-Characteristic Area Under the Curve with the False Positive Rate in [0.001, 0.1] (**logAUC$_{[0.001, 0.1]}$**) [22]: This is used because only a small percentage of molecules predicted with high activity can be selected for experimental tests in consideration of cost in a real-world drug campaign [19]. Receiver-Operating-Characteristic Area Under the Curve (**AUC**): AUC is included since it has historically been used as a general purpose evaluation metric for graph classification [23].

## 4.2 Experimental Results

From Table 1, we can see MolKGNN achieves superior results in recovering the active molecules with a high decision threshold. This highlights the ability of the proposed model to perform well in the application-related metric. Moreover, we find MolKGNN also performs on par with other GNN in terms of AUC, which demonstrates its applicability beyond drug discovery in a general setting. It is worth noting that different ranking of models are observed in the two tables. This demonstrates that a generally good performing model measured by AUC could potentially perform bad in a specific false positive rate region. Moreover, the learned kernel shown in Figure 2 (A) reveals a pattern of a center atom of carbon surrounded by neighboring three fluorine and another carbon. This pattern is known as the trifluoromethyl group in medicinal chemistry and has been used in several drugs [24]. The details of interpretability can be found in Appendix A.8. We also perform an additional experiment to exhibit MolKGNN's ability to distinguish chirality in Appendix A.5.

**Ablation Studies.** *Component of $\phi(S, S')$*: Results show in Figure 2 (B). *Kernel Number*: Results show in Figure 2 (C). We provide a discussion on these results in Appendix A.9.

# 5 Conclusion

We introduce a new GNN model named MolKGNN to address the QSAR model construction for CADD. MolKGNN utilizes a newly-designed molecular convolution, where a molecular neighbor-

**Table 1:** Results on the domain-related metric, logAUC$_{[0.001,0.1]}$ (summarized the general metric, AUC - details in Appendix A.7) over five runs. It shows that a well-performing model measured by a general metric could potentially perform badly in the application-related metric.

| PubChem AID | MolKGNN (ours) | SchNet | SphereNet | DimeNet++ | ChiRo | KerGNN |
|---|---|---|---|---|---|---|
| 435008 | $0.255 \pm 0.014$ | $0.187 \pm 0.027$ | $0.215 \pm 0.024$ | $0.203 \pm 0.047$ | $0.168 \pm 0.019$ | $0.147 \pm 0.015$ |
| 1798 | $0.174 \pm 0.029$ | $0.195 \pm 0.025$ | $0.196 \pm 0.035$ | $0.208 \pm 0.035$ | $0.165 \pm 0.040$ | $0.078 \pm 0.042$ |
| 435034 | $0.227 \pm 0.022$ | $0.246 \pm 0.020$ | $0.230 \pm 0.034$ | $0.235 \pm 0.044$ | $0.211 \pm 0.023$ | $0.179 \pm 0.045$ |
| 1843 | $0.362 \pm 0.033$ | $0.358 \pm 0.037$ | $0.258 \pm 0.048$ | $0.284 \pm 0.034$ | $0.326 \pm 0.010$ | $0.292 \pm 0.027$ |
| 2258 | $0.301 \pm 0.028$ | $0.240 \pm 0.037$ | $0.380 \pm 0.037$ | $0.340 \pm 0.032$ | $0.251 \pm 0.010$ | $0.195 \pm 0.020$ |
| 463087 | $0.390 \pm 0.056$ | $0.332 \pm 0.022$ | $0.399 \pm 0.011$ | $0.389 \pm 0.026$ | $0.258 \pm 0.019$ | $0.150 \pm 0.011$ |
| 488997 | $0.303 \pm 0.027$ | $0.319 \pm 0.017$ | $0.309 \pm 0.029$ | $0.315 \pm 0.011$ | $0.193 \pm 0.029$ | $0.081 \pm 0.023$ |
| 2689 | $0.415 \pm 0.020$ | $0.324 \pm 0.020$ | $0.401 \pm 0.016$ | $0.367 \pm 0.049$ | $0.351 \pm 0.048$ | $0.264 \pm 0.017$ |
| 485290 | $0.498 \pm 0.015$ | $0.333 \pm 0.047$ | $0.450 \pm 0.039$ | $0.463 \pm 0.040$ | $0.295 \pm 0.068$ | $0.223 \pm 0.026$ |
| Average | 0.325 | 0.282 | 0.315 | 0.312 | 0.247 | 0.179 |
| Avg. Rank | **2.333** | 3.222 | 2.556 | 2.556 | 4.556 | 5.778 |
| AUC Average | 0.843 | 0.844 | 0.826 | 0.823 | 0.823 | 0.816 |
| AUC Avg. Rank | 2.889 | **2.111** | 3.778 | 3.889 | 4.000 | 4.222 |

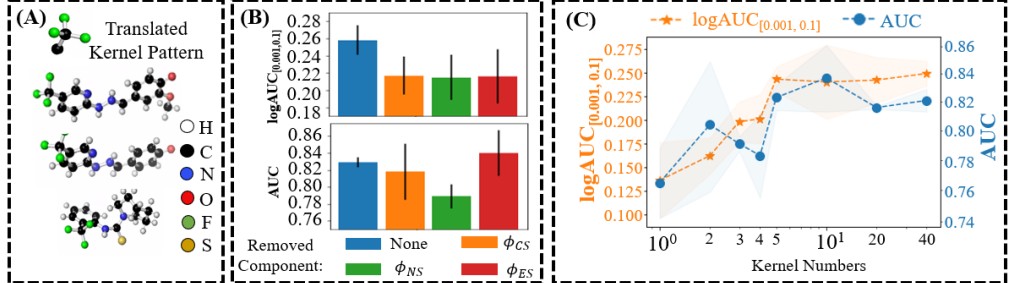

**Figure 2:** (A) Visualization of a learned kernel and three examples. (B) Ablation study result for $\phi(S, S')$ components. (C) Performance for different kernel numbers.

hood is compared with a molecular kernel to output a similarity score. Comprehensive benchmarking is conducted to evaluate MolKGNN to show its superiority over existing GNN baselines.

# 6 Acknowledgements

This work was supported through NIDA R01 DA046138. JM acknowledges funding by the Deutsche Forschungsgemeinschaft (DFG, German Research Foundation) through SFB1423 and SPP 2363. JM is supported by a Humboldt Professorship of the Alexander von Humboldt Foundation. BB acknowledges funding by the National Science Foundation under grant 1763966. YL acknowledges the support from the NVIDIA Academic Hardware grant.

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

## A Appendix

### A.1 Datasets

PubChem [25] is a database supported by National Institute of Health (NIH) that contains biological activities for millions of drug-like molecules, often from HTS experiments. However, the raw primary screening data from PubChem have a high false positive rate [19, 20]. We benchmark our model using nine high-quality HTS experiments from PubChem that cover all important protein classes for drug discovery [19, 20] (statistics in Table 2 where each dataset was carefully curated to have lists of inactive and confirmed active molecules from secondary experimental screens).

### A.2 Experiment Details

**Data Preprocessing** We preprocessed the input SMILES strings to Structure-Data Files (SDFs). Each dataset is specified by its PubChem BioAssay accession (AID) [26]. Prepossessing to the original

**Table 2:** Statistics of datasets used in the experiment. The datasets feature in the large data size, highly imbalanced labels, and diverse protein targets. Datasets are identified by their PubChem Assay ID (AID).

| Protein Target Class | Protein Target (PubChem AID) | Total # of Graphs | # of Active Labels | Per Graph Avg. # of Nodes (Edges) |
|---|---|---|---|---|
| GPCR | Orexin1 Receptor (435008) | 218,156 | 233 | 45.14 (94.37) |
| | M1 Muscarinic Receptor Agonists (1798) | 61,832 | 187 | 43.60 (91.37) |
| | M1 Muscarinic Receptor Antagonists (435034) | 61,755 | 362 | 43.61 (91.41) |
| Ion Channel | Potassium Ion Channel Kir2.1 (1843) | 301,490 | 172 | 44.41 (92.81) |
| | KCNQ2 Potassium Channel (2258) | 302,402 | 213 | 44.44 (92.88) |
| | Cav3 T-type Calcium Channels (463087) | 100,874 | 703 | 43.75 (91.57) |
| Transporter | Choline Transporter (488997) | 302,303 | 252 | 44.46 (92.90) |
| Kinase | Serine/Threonine Kinase 33 (2689) | 319,789 | 172 | 44.85 (93.70) |
| Enzyme | Tyrosyl-DNA Phosphodiesterase (485290) | 341,304 | 281 | 46.13 (96.50) |

data includes converting SMILES strings to 3D SDF files, generating 3D conformation, and filtering. Conversion from SMILES to SDF files is done using Open Babel [27], version 2.4.1. Conformations are generated using Corina [28], version 4.3. Molecules are further filtered with validity, duplicates with BioChemical Library (BCL) [29].

**Training Details** The datasets are randomly split into 80%/10%/10% for training, validation, and testing respectively. We then shrink the training set to contain only 10,000 inactive-labeled molecules, while keeping all active-labeled molecules. This shrinking technique was previously used by [30]

By shrinking the training data size, we can shorten the training time given the limited computational resources, while keeping most active signal that we're interested in. We did an empirical study on the shrinking effect on AID 2258 (302,402 molecules). Results are shown in Table 3. We can see there is indeed a decrease of performance in terms of logAUC$_{[0.001,0.1]}$. We leave the benchmarking of the full datasets in a future study.

To overcome the highly-imbalanced problem, we sample the training data in each batch according to the inverse frequency of the label occurrence in the training set. For example, if the active label appear at 1% rate in the training set, it has a sampling weight of 1/0.01 =100, while if the inactive label appear 99% of time in the training set, it gets a sampling weight of $1/0.99 \approx 1.01$. The active-labeled data are thus roughly 100 times more likely to be sampled than inactive-labeled data in each batch.

During the training, in the forward propagation, the kernels are convoluted with the atoms having the same degree. In the backward propagation, the parameters with degree d are only updated if the molecule has nodes of degree d. AdamW optimizer is used [31] for the training. The codes are implemented using PyTorch [32] and PyG [33].

**Hyperparameters Search Space** See Table 4 for details.

**Hyperparameters** See Tables 5 for details for training MolKGNN. For other benchmarking models except KerGNN, we use the same hyperparameters from their codes.

For KerGNN, we empirically observe that using the default hyperparameter setting achieves significantly low performance on our well-curated datasets and hence we further tune its hyperparamters as follows: batch size $\{64, 128\}$, the hidden unit of the linear layer $\{16, 32\}$.

## A.3 Featurization

Different models have different ways of featurization. We use the original features reported in the original papers for each model used in the benchmarking. Our featurization is adapted from [10]. Rdkit(version 2022.3.4) [34] is used for the featurization. See Table 6 and 7 for details.

**Table 3:** Comparing sample and full training results on AID 2258 using MolKGNN over three runs.

| Inactive Training Size | logAUC$_{[0.001,0.1]}$ | AUC |
|---|---|---|
| 10K Sample | $0.296 \pm 0.026$ | $0.820 \pm 0.021$ |
| Full | $0.384 \pm 0.003$ | $0.816 \pm 0.030$ |

**Table 4:** Hyperparameter search space used for MolKGNN.

| Hyperparameter | Search Space |
|---|---|
| Hidden Dimension | {32, 64} |
| Batch Size | {16, 32} |
| # Layers | {1, 2, 3, 4, 5} |
| Peak Learning Rate | {5e-1, 5e-2, 5e-3, 5e-4} |
| Dropout | {0.1, 0.2, 0.3} |

**Table 5:** Hyperparameters used for MolKGNN.

| Hyperparameter | Value |
|---|---|
| Node Feature Dimension | 28 |
| Edge Feature Dimension | 7 |
| Hidden Dimension | 32 |
| Batch Size | 16 |
| # Layers | 4 |
| # of Kernels of Degree 1 | 10 |
| # of Kernels of Degree 2 | 20 |
| # of Kernels of Degree 3 | 30 |
| # of Kernels of Degree 4 | 50 |
| Warmup Steps | 300 |
| Peak Learning Rate | 5e-3 |
| End Learning Rate | 1e-10 |
| Weight Decay | 0.001 |
| Epochs | 20 |
| Dropout | 0.2 |

**Table 6:** Node features $\mathbf{X}_v$ for $v$

| Indices | Description |
|---|---|
| 0-11 | One-hot encoding of element type: H, C, N, O, F, Si, P, S, Cl, Br, I, other |
| 12-15 | One-hot encoding of node degree: 1, 2, 3, 4 |
| 16 | Formal charge |
| 17 | Is in a ring |
| 18 | Is aromatic |
| 19 | Explicit valence |
| 20 | Atom mass |
| 21 | Gasteiger charge |
| 22 | Gasteiger H charge |
| 23 | Crippen contribution to logP |
| 24 | Crippen contribution to molar refractivity |
| 25 | Total polar surface area contribution |
| 26 | Labute approximate surface area contribution |
| 27 | EState index |

**Table 7:** Edge features $\mathbf{E}_{vu}$ for $e_{vu}$

| Indices | Description |
|---|---|
| 0 | Is aromatic |
| 1 | Is conjugate |
| 2 | Is in a ring |
| 3-6 | One-hot encoding of bond type: 1, 1.5, 2, 3 |

## A.4  2.5D vs 3D

While many previous work have attempted to develop 3D models by including distance, angles, torsions into their model designs [6, 11, 13], we demonstrated that 2.5D model can achieve comparable results in terms of AUC. We provide the explanation of why a model with seemly less information can accomplish this from a chemistry perspective. The bond lengths/angles have little variations given the certain involving atom identities and bond types [35, 36]. Moreover, different than determining bond lengths/angles experimentally, many programs such as Corina [28] that converts SMILES to 3D SDF using standard bond lengths/angles [1], which stay the same in different molecules. Hence bond lengths/angles provide little additional information in distinguishing different molecules. This can also been seen by the fact that an experienced chemist can just look at a molecular structure and know certain properties of the molecule, without the need to know the exact bond lengths/angles. Nevertheless since our model has the potential to integrate bond length and angles into the $\phi(S, S')$, we plan to include those for comparison in the future studies.

On the other hand, molecules can have different conformations as a result of the single bond rotation. The same molecule with different conformation consequently has different sets of torsions. However, the pharmacological activity is usually linked with few conformations (binding conformation) and hence related to certain sets torsions. It seems that knowing torsion could potentially be help the activity prediction. Nevertheless, knowing which conformation is the binding conformation is a challenging task. A set of torsions related with a wrong predicted binding conformation is detrimental to the model performance. Hence we decide to build a conformation-invariant model and exclude torsion to circumvent this problem.

## A.5  Ability to Distinguish Chirality

We further experiment on the expressiveness of our model to determine whether it is able to distinguish chiral molecules. We use the CHIRAL1 dataset [8] that contains 102,389 enantiomer pairs for a single 1,3-dicyclohexylpropane skeletal scaffold with one chiral center. The data is labeled as R or S stereocenter and we use accuracy to evaluate the performance. For comparison, we use GCN [37] and a modified version of our model, MolKGNN-NoChi, that removes the chirality calculation module. Our experiments observed GCN and MolKGNN-NoChi achieve 50% accuracy while MolKGNN achieves nearly 100%, which empirically demonstrates our proposed method's ability to distinguish chiral molecules.

## A.6  Computation Complexity

It may seem to be formidable to enumerate all possible matchings described in Section 3.1. However, most nodes only have one neighbor (e.g., hydrogen, fluorine, chlorine, bromine and iodine). Take AID 1798 for example, 49.03%, 6.12%, 31.08% and 13.77% nodes are with one, two, three and four neighbors among all nodes, respectively. For nodes with four neighbors, only 12 out of 24 matchings need to be enumerated because of chirality [8].

The computation complexity incurs from two operations: (1) kernel convolution and (2) message propagation. In the kernel convolution, as analyzed above, the permutation is bounded by up to four neighbors (12 matchings). Hence finding the optimal matching takes $\mathcal{O}(1)$ per node per kernel and $\mathcal{O}(|\mathcal{V}|K)$ for the whole graph. In the message propagation, each edge passes the $K$-dimensional feature from its head to its tail node, taking $\mathcal{O}(K)$ time, and since we have $|\mathcal{E}|$ edges in total, the message propagation takes $\mathcal{O}(|\mathcal{E}|K)$ time. Therefore, the total time complexity of the above two operations is $\mathcal{O}((|\mathcal{V}| + |\mathcal{E}|)K)$, which is asymptotically equivalence to the time complexity of many existing graph convolutions considering no feature transformation and assuming the feature dimension there is also $K$.

## A.7  AUC Result

See Table 8 for details.

---

[1]This is explicitly mentioned in: https://mn-am.com/wp-content/uploads/2021/10/corina_classic_manual.pdf

**Table 8:** Comparison of AUC between models. The performance is better when the value is higher. Reported are the mean values over five runs, with standard deviation.

| PubChem AID | MolKGNN (ours) | SchNet | SphereNet | DimeNet++ | ChiRo | KerGNN |
|---|---|---|---|---|---|---|
| 435008 | $0.836 \pm 0.012$ | $0.820 \pm 0.009$ | $0.794 \pm 0.026$ | $0.787 \pm 0.028$ | $0.797 \pm 0.015$ | $0.806 \pm 0.017$ |
| 1798 | $0.721 \pm 0.027$ | $0.707 \pm 0.007$ | $0.655 \pm 0.025$ | $0.649 \pm 0.028$ | $0.683 \pm 0.052$ | $0.663 \pm 0.041$ |
| 435034 | $0.816 \pm 0.028$ | $0.838 \pm 0.009$ | $0.836 \pm 0.014$ | $0.834 \pm 0.019$ | $0.822 \pm 0.017$ | $0.821 \pm 0.016$ |
| 1843 | $0.879 \pm 0.025$ | $0.896 \pm 0.012$ | $0.875 \pm 0.021$ | $0.857 \pm 0.011$ | $0.881 \pm 0.010$ | $0.906 \pm 0.020$ |
| 2258 | $0.806 \pm 0.019$ | $0.792 \pm 0.020$ | $0.801 \pm 0.042$ | $0.821 \pm 0.025$ | $0.782 \pm 0.018$ | $0.766 \pm 0.024$ |
| 463087 | $0.895 \pm 0.003$ | $0.910 \pm 0.005$ | $0.904 \pm 0.005$ | $0.902 \pm 0.009$ | $0.891 \pm 0.004$ | $0.859 \pm 0.009$ |
| 488997 | $0.866 \pm 0.018$ | $0.831 \pm 0.012$ | $0.822 \pm 0.017$ | $0.839 \pm 0.023$ | $0.817 \pm 0.019$ | $0.757 \pm 0.044$ |
| 2689 | $0.906 \pm 0.019$ | $0.905 \pm 0.021$ | $0.867 \pm 0.021$ | $0.832 \pm 0.016$ | $0.919 \pm 0.017$ | $0.912 \pm 0.013$ |
| 485290 | $0.866 \pm 0.012$ | $0.893 \pm 0.011$ | $0.879 \pm 0.021$ | $0.884 \pm 0.016$ | $0.816 \pm 0.015$ | $0.853 \pm 0.009$ |
| Avgerage | 0.843 | 0.844 | 0.826 | 0.823 | 0.823 | 0.816 |
| Avg. Rank | 2.889 | **2.111** | 3.778 | 3.889 | 4.000 | 4.222 |

## A.8 Investigation of Interpretability

Because the atom features are transformed into a node embedding in the MolKGNN via batch normalization, the learned kernels are also in this node embedding space, which is not directly human-readable.

To intepret the contents in kernels, we train an interpreting model to convert the node embedding in the learnable kernels back into human-readable atomic number that represents the element type. The interpreting model is of an autoencoder-like architecture, that contains an encoder and a decoder. The encoder is architecturally the same as the one in MolKGNN, which is a batch normalization. It takes in the atom embedding and outputs a node embedding. The decoder takes in the node embedding and outputs the human-readable atomic number.

The training dataset is AID 1798. The input is the atom features from molecules in AID 1798. The ground truth is the atomic number of the atom, which can be extracted from the first 12 digits in the features (See Table 6). After the training, the decoder acquires the ability to convert a node embedding back to the atomic number.

Finally, this encoder can be used to translate the node embedding in the kernels into atomic numbers. We examine the learned kernels and Figure 2 (A) is one example that demonstrates the interpretability of our model from dataset AID 2689. Currently we only examine the first layer and the node attributes of the kernels, but our kernels offer the potentials for retrieving more complicated pattern and we leave the investigation of that for future works.

## A.9 Ablation Study Details

From the result in Figure 2 (B) shows that the removal of any of the components has a negative impact on $\text{logAUC}_{[0.001,0.1]}$. In fact, the impact is bigger for $\text{logAUC}_{[0.001,0.1]}$ than AUC in terms of the percentage of performance change. Note that in some cases such as the removal of $\phi_{\text{es}}$, there is an increase in performance according to AUC, but this would significantly hinder the $\text{logAUC}_{[0.001,0.1]}$ metric.

Results in Figure 2 (C) shows that when the number of kernels is too small ($< 5$), it greatly impacts the performance. However, once it is large enough to a certain point, a larger number of kernels has little impact on the performance.

The ablation studies are conducted using dataset AID 435008. Reported are average values over three runs, with standard deviation. The number of kernels shown in Figure 2 (C) is the number of kernels per degree, instead of total number of kernels.

## A.10 Evaluation Metrics Details

- Logarithmic Receiver-Operating-Characteristic Area Under the Curve with the False Positive Rate in [0.001, 0.1] (**logAUC$_{[0.001,0.1]}$**): Ranged logAUC [22] is used because only a small percentage of molecules predicted with high activity can be selected for experimental tests in consideration of cost in a real-world drug campaign [19]. This high decision cutoff corresponds to the left side of the Receiver-Operating-Characteristic (ROC) curve, i.e., those False Positive Rates (FPRs) with small values. Also, because the threshold cannot be predetermined, the area under the curve is used to consolidate all possible thresholds within a certain FPR range. Finally, the logarithm is used to bias

towards smaller FPRs. Following prior work [30, 38], we choose to use $\text{logAUC}_{[0.001,0.1]}$. A perfect classifier achieves a $\text{logAUC}_{[0.001,0.1]}$ of 1, while a random classifier reaches a $\text{logAUC}_{[0.001,0.1]}$ of around 0.0215, as shown below:

$$\frac{\int_{0.001}^{0.1} x \, \mathrm{d} \log_{10} x}{\int_{0.001}^{0.1} 1 \, \mathrm{d} \log_{10} x} = \frac{\int_{-3}^{-1} 10^u \mathrm{d}u}{\int_{-3}^{-1} 1 \mathrm{d}u} \approx 0.0215$$

$$0.0215 \left( \frac{\int_{0.001}^{0.1} x \, \mathrm{d} \log_{10} x}{\int_{0.001}^{0.1} 1 \, \mathrm{d} \log_{10} x} = \frac{\int_{-3}^{-1} 10^u \mathrm{d}u}{\int_{-3}^{-1} 1 \mathrm{d}u} \approx 0.0215 \right).$$

- Receiver-Operating-Characteristic Area Under the Curve (**AUC**): We include AUC since this has historically been used as a general purpose evaluation metric for graph classification [23]. Comparison with AUC also highlights the fact that overall performance (ranking) of methods according to AUC may not align well with that of the domain specific evaluation metric, i.e., $\text{logAUC}_{[0.001,0.1]}$. Receiver-Operating-Characteristic Area Under the Curve (AUC)

  Plain AUC is included here to benchmark the methods' performance for general purposes. It also serves as a comparison to the $\text{logAUC}_{[0.001,0.1]}$ to highlight the fact that the best general good performing may classifier may not be the best at a high threshold.

## A.11  Baseline Details

**SchNet** [6] is one of the early attempts to extend convolution to molecular representation learning. The traditional convolution can only be applied to grid-like data such as images using discrete filters. This work proposes continuous-filter convolutional layers to be able to model local correlations without requiring the data to lie on a grid.

**DimeNet++** [21] builds on top on DimeNet [11], which resembles belief propagation. It integrates bond length and angles information into the message passing step by using spherical Bessel functions and spherical harmonics.

**SphereNet** [13] proposes a spherical message passing (SMP) to include atom 3D coordinates. SMP captures relative atom position in the spherical coordinate system and hence enables the chirality characterization.

**ChIRo** [7] designs a novel torsion encoder that is invariant to bond rotation, while being able to learn molecular chirality. This torsion encoder leverage the factor that rotating a bond will change coupled torsions together to achive the conformation-invariance. A phase shift is added to the torsion encoder to break the chirality symmertry.

**KerGNN** [5] is different from the above four models that are specifically designed for molecular representation learning. KerGNN is architecturally similar to ours in the fact that it quantifies the similarity between a subgraph with a kernel via graph kernel method. However, we argue that this structural similarity is not as helpful as the semantic similarity in molecular representation learning tasks. This argument is verified by the experiment in Section 4.

