# OpenReview forum: "Interpretable Chirality-Aware Graph Neural Network for Quantitative Structure Activity Relationship Modeling"
_logconference.io/LOG/2022/Conference — LoG 2022 Poster_

### Official Review · Reviewer_9Zjn · 2022-10-17

**Overall Score:** 8
**Confidence:** 3

**Review:**

**Summary**:
This paper proposes a kernel-based graph neural network for molecular property prediction that takes chirality into account. In my opinion, the most significant contribution of this paper is the chirality accounted graph kernel, which is verified by the results in A.5. Though there are some problems in results reporting and the paper writing, I think they are not fatal and recommend accepting it with some revision.

**Pro**:
- This paper is tackling an important but underestimated problem: chirality. Discussion in A.4 is well-written on this topic, and I recommend moving it to the main text if the authors are allowed to submit with extra pages.
- The explicit adaption of the kernel does add interpretability to the model, though it can hardly be illustrated. However, this is not the problem of this paper, and I still think this is a plus point to the paper.

**Con**:
- In Figure 2B, the performance after removing $\phi_{ES}$ seems better than the one without ablation in the top figure. However, it seems different in the bottom one. I am not sure why the logarithm of AUC is needed as a metric, and I think Figure 2B needs more explanation and annotation.
- Lack of novelty. The idea of graph kernel is not novel, and the form of chirality seems like from [1]. Overall, the technical contribution is relatively limited.

**Other Comments**:
- A more detailed explanation about how the kernels are learned is desired.
- Table 1 has two extra lines without AUC, which should be removed.
- In line 85, the Figure’s number is missing.

Reference:
[1] Pattanaik, L., Ganea, O. E., Coley, I., Jensen, K. F., Green, W. H., & Coley, C. W. (2020). Message passing networks for molecules with tetrahedral chirality. arXiv preprint arXiv:2012.00094.

---

### Official Review · Reviewer_WvX9 · 2022-10-20

**Overall Score:** 8
**Confidence:** 4

**Review:**

**Summary**

The paper proposes MolKGNN, a GNN with learnable kernels/sub-structure, where chirality differences between molecules and kernels are also considered. The authors show great model performance on HTS experiments.

**Strength**

- The idea of matching learnable kernels is clean and intuitive. Individual components of the methods are also well-motivated and -described.
- The experiments are well throughout, with carefully selected datasets, metrics, and ablation studies showing the benefit of different module components.
- The proposed method shows strong performance against some very strong baselines compared to many of the strong baselines in today's GNN literature.

**Weakness**

- While the authors might select HTS datasets to showcase the necessity of chirality for the binding model, the chirality is only part of the features. I am curious to see whether MolKGNN could be a general GNN that works well on other molecular property tasks, such as those in MolNet or OGB.
- I am not convinced by "interpretability through anecdotal examples", so I encourage the authors to think about a systematic measurement for attribution. Evaluating Attribution for Graph Neural Networks by Sanchez-Lengeling et al. and the follow-up literature might be helpful.
- There are too many minor details things in the paper. To name a few:
1) missing figure reference at line 85
2) AUC and logAUC averages appear to be the same in Table 1
3) missing x-axis label in Figure 2C
4) typo in line 356 in the appendix

**Recommendation**

The paper proposes an interesting GNN architecture, especially for solving chirality, and shows great performance through a benchmark. I am giving a weak accept (6) for the paper in its current form. I **will** raise my score to clear accept (8) once the authors address the minor details I pointed out above, and **might consider** raise to strong accept (10) if the authors can address my first two points in the weakness section.

---

### Official Review · Reviewer_8Pi7 · 2022-10-20

**Overall Score:** 8
**Confidence:** 3

**Review:**

**Summary**

This paper introduces a new molecule representation learning framework MolKGNN, which is chirality aware, interpretable and achieves reasonable performance on structure activity prediction tasks. Formally, given a graph G, the task is to predict a property y(G).

Instead of resorting to the usual message-passing framework, the paper builds on recent advances in differentiable graph kernels (or filters) and introduce additional components capturing edge feature similarities and chirality into the learning setup. The learnable kernels themselves consist of node and edge features based on a 1-hop neighbourhood for different degree.

The model is evaluated on different high throughput screening (HTS) tasks across major protein classes and achieves reasonable performance.

**Pros**:

1. The paper is very well written, easy to follow and has appropriate references to prior work.

2. The baselines compared to are among the current SoTA methods, and the authors put in great effort to run these baselines multiple times across the different tasks, and tune hyperparameters.

3. The appendix also has a lot of essential details and sanity checks (regarding chirality for ex) that lend further credence to the author's efforts.

**Potential Cons / Concerns**:

Overall, I do not have any strong concerns with this work. There are some points that can receive attention for future work:

1. Interpretability is highlighted as one of the key objectives of this work, but some of this interpretability could be hampered by restriction to the star-like 1-hop neighborhoods, especially if the substructures contributing to an activity have multiple rings.

One potential way of addressing this could be to consider decomposing the molecule into motifs, and learning graph kernels on this motif-based meta graph.

2. It is a bit interesting that for AUC, removing the edge feature similarity calculation improves the performance. Did the authors look into this, and could possibly shed some light?

3. In the appendix section on computational complexity, it is said that the cost of optimal matching is O(1). Shouldn't this be O(n) where n is the number of nodes in the graph - (there seems to be a 0.25 factor between the number of 1 and 4 degree atoms). I might be mistaken here, and any clarification in this regard is appreciated.

4. The edge matching pipeline could be replaced with an optimal transport equivalent [1]

**Typos**:

1. Reference to figure is missing in Line 85

2. SMILES instead of SMIELS on line 269

**Reasons for Score**:

Overall I lean towards Accept for this paper. The paper is well written, makes sound modelling choices with technical novelty and achieves good experimental performance. Further experiments in the appendix support the claims in the Appendix.

[1] Cuturi et al. Sinkhorn Distances: Lightspeed Computation of Optimal Transport (2013)

---

### Official Review · Reviewer_6Ao1 · 2022-10-20

**Overall Score:** 5
**Confidence:** 4

**Review:**

This manuscript proposes a new graph neural network model for molecular representation learning. The paper introduces some high-level patterns of the atom graph to have better representation in the downstream tasks.
Major:
1) The major contribution of the paper is chirality-aware, but how the three similarities in the proposed model are linked to chirality is not clear in the text.
2) The goal of the experiments is not clear, with the developed model, what is the goal of the model training? The binary prediction on which characteristics of the molecule? To predict R or S?
3) The interpretation part is also not clear, like Figure 2A, what does the color mean and how the interpretation part works quantitatively?
4) It looks like the novelty of the proposed model is mainly on the three similarities, and this high-order structure of the niche is vital in the performance. This motif-like structure in GNN is not new. It may be interesting to illustrate the contributions of these similarities in molecule studies quantitatively. That may be much more interesting and make the model interpretable.

Minor:
Line 85, the figure is not linked correctly with ??

---

### Meta-Review · Area_Chair_B6wd · 2022-11-15

**Confidence:** 4
**Recommendation:** Accept

**Meta Review:**

This work aims to build a chirality-aware graph neural network for drug discovery. It is an important problem in drug discovery but understudied in the field of machine learning. This paper presents a new way of representing chirality via learnable, interpretable kernels and demonstrates reasonable empirical improvement. During the discussion stage, three out of four reviewers raised their score to 8 based on authors' response. While the other reviewer (6Ao1) still remains weak reject, I believe the modified manuscript is ready for acceptance based on other three reviewers' judgment.

---

### Decision · Program_Chairs · 2022-11-22

Accept (Poster)